# Analysis of low-density lipoprotein receptor gene mutations in a family with familial hypercholesterolemia

Ya-nan Hu[1☯], Min Wu[1☯], Hong-ping Yu[1☯], Qiu-yan Wu[1☯], Ying Chen[1,2☯], Jian-Hui Zhang[1‡], Dan-dan Ruan[1‡], Yan-ping Zhang[1‡], Jing Zou[1‡], Li Zhang[1,3‡], Xin-fu Lin[1,4‡], Zhu-ting Fang[1,5‡], Li-Sheng Liao[1,6]*, Fan Lin[1,7]*, Hong Li[1,2]*, Jie-Wei Luo[1,2]*

1 Department of Traditional Chinese Medicine, Shengli Clinical Medical College of Fujian Medical University, Fujian Provincial Hospital, Fuzhou, China, 2 Department of Traditional Chinese Medicine, Fujian Provincial Hospital, Fuzhou, China, 3 Department of Nephrology, Fuzhou University Affiliated Provincial Hospital, Fuzhou, China, 4 Pediatrics department, Fujian Provincial Hospital, Fuzhou, China, 5 Department of Oncology and Vascular Intervention, Fujian Provincial Hospital, Fuzhou, China, 6 Department of Hematology, Fujian Provincial Hospital, Fuzhou, China, 7 Department of Geriatric Medicine, Fujian Provincial Center for Geriatrics, Fujian Provincial Hospital, Fuzhou, China

☯ These authors contributed equally to this work.
‡ These authors also contributed equally to this work.
* liaolisheng@126.com (L-SL); linfan@fjmu.edu.cn (FL); 913836396@qq.com (HL); docluo0421@aliyun.com (J-WL)

**Data Availability Statement:** "The data underlying the results presented in the study are available from National Center for Biotechnology

## Abstract

### Background

Familial hypercholesterolemia (FH) is a common monogenic autosomal dominant disorder, primarily mainly caused by pathogenic mutations in the low-density lipoprotein receptor (LDLR) gene. Through phenotypic-genetic linkage analysis, two *LDLR* pathogenic mutations were identified in FH families: c.G1027A (p.Gly343Ser) and c.G1879A (p.Ala627Thr).

### Materials and methods

Whole exome sequencing was conducted on the proband with familial hypercholesterolemia to identify the target gene and screen for potential pathogenic mutations. The suspicious responsible mutation sites in 14 family members were analyzed using Sanger sequencing to assess genotype-phenotype correlations. Mutant and wild type plasmids were constructed and transfected into HEK293T cells to evaluate LDLR mRNA and protein expression. In parallel, bioinformatics tools were employed to predict structural and functional changes in the mutant LDLR.

### Results

Immunofluorescence analysis revealed no significant difference in the intracellular localization of the p.Gly343Ser mutation, whereas protein expression of the p.Ala627Thr mutation was decreased and predominantly localized in the cytoplasm. Western blotting has showed that protein expression levels of the mutant variants were markedly declined in both cell lysates and supernatants. Enzyme linked immunosorbent assay has demonstrated that

Information: SCV002098042 (https://www.ncbi.nlm.nih.gov/clinvar/variation/183106/); SCV002098051 (https://www.ncbi.nlm.nih.gov/clinvar/variation/252101/)."

**Funding:** This work was supported by the Fujian Province Natural Science Fund Project (2022J01996, 2022J01409, 2021J02053, 2023J011159), Fujian Provincial Youth Scientific Program on health (2021QNB001), the Fujian Province Medical Innovation Foundation (2022CXB002, 2022CXA001), the Special Research Foundation of Fujian Provincial Department of Finance (No. 2021-848, 2021-917, 2022-840), and National famous and old Chinese medicine experts (Xuemei Zhang, Xiaohua Yan, Shaoguang Lv) inheritance studio construction project.

**Competing interests:** The authors have declared that no competing interests exist.

**Abbreviations:** FH, Familial hypercholesterolemia; LDLR, low-density lipoprotein receptor; LDL-C, LDL cholesterol; TC, total cholesterol; ARH, autosomal recessive hypercholesterolemia; ASCVD, atherosclerotic cardiovascular disease; PCSK9, proprotein convertase subtilisin/kexin type 9; APOB, apolipoprotein B; LDLRAP1, Low-density lipoprotein receptor adaptor protein 1; HeFH, heterozygous FH; HoFH, homozygous FH; TG, triglyceride; HDL-C, high-density lipoprotein cholesterol; apoA, apolipoprotein A; CTA, computed tomography angiography; SNPs, single nucleotide polymorphisms; ACMG, American College of Medical Genetics and Genomics; CAD, coronary artery disease; ER, endoplasmic reticulum; ABCG, ATP-binding cassette sub-family G member.

LDLR protein levels in the supernatant of cell culture medium was not significant different from those of the wild-type group. However, LDLR protein levels in the cell lysate of both the Gly343Ser and Ala627Thr variants groups were significantly lower than those in the wild-type group. Bioinformatic predictions further suggested that these mutations may affect post-translational modifications of the protein, providing additional insight into the mechanisms underlying the observed reduction in protein expression.

## Conclusions

In this study, we identified two heterozygous pathogenic variants in the *LDLR* gene, c. G1027A (p.Gly343Ser) and c.G1879A (p.Ala627Thr), in a family with familial hypercholesterolemia. We also conducted preliminary investigations into the mechanisms by which these mutations contribute to disease pathology.

## Introduction

Familial hypercholesterolemia (FH) is a common autosomal genetic disorder, predominantly inherited in a dominant manner, with the exception of the rare autosomal recessive variant, ARH [1]. The prevalence of heterozygous FH is estimated to be 1 in 200–300 individual, while homozygous FH occurs in approximately 1 in 170,000–300,000 [2]. The primary etiological factor of FH involves mutations in the *LDLR* gene, manifesting in acute elevations of total and LDL cholesterol levels. This genetic disorder often presents with diverse phenotypic expressions, includingxanthomas, cardiac valve abnormalities, corneal arcus, and an elevated risk of early-onset coronary artery disease [3–6]. Individuals with FH are subjected to substantially elevated LDL-C levels from birth, significantly heightening their risk for atherosclerotic cardiovascular diseases (ASCVD) including myocardial infarction. The risk of premature coronary artery disease in FH patients is approximately 20 times higher than in the general population [7,8].

The LDL receptor (LDLR), Proprotein Convertase Subtilisin/Kexin Type 9 (PCSK9), and Apolipoprotein B (APOB) are the primary genes implicated in the autosomal dominant form of FH [9,10]. In contrast, autosomal recessive hypercholesterolemia (ARH) involves distinct molecular pathways, often related to LDL receptor adaptor protein 1 (LDLRAP1) [11]. Among these, the *LDLR* gene is the most frequently mutated, accounting for approximately 93% of all genetic alterations associated with FH [12]. The seminal discovery by Goldstein and Brown in 1974 emphasized that mutations in the *LDLR* gene impaired LDL-C binding to its receptor, thereby disrupting cholesterol homeostasis and precipitating severe atherosclerosis from an early age [13]. The comprehensive utilization of advanced genetic diagnostics and the enhancement of dedicated FH registries are crucial for the proactive management and prevention of FH.

In this study, we investigated a FH family with coronary heart disease and cutaneous xanthomatosis caused by compound heterozygous mutations in the *LDLR* gene. These mutations were detected using whole exome sequencing. Functional studies in vitro validated the impact of these mutations on cellular and molecular function and examined their intracellular locations. We also conducted a preliminary analysis of the association between phenotypic traits and gene mutations. Additionally, bioinformatics tools were employed to predict the structure and function of the mutant protein, providing valuable insights for future research.

## Methods

### Research subjects

The proband, a 53-year-old Han Chinese woman from Fujian, China, presented with a 20-year history of recurrent chest tightness. A soybean-sized xanthoma was observed on her left upper eyelid, accompanied by a corneal arcus. Plasma lipid tests repeatedly showed significantly elevated levels, and coronary CTA revealed eccentric mixed plaques in the proximal left main artery, extending to the anterior descending branch, with stenosis of 70%–80%. A similar medical history was noted in her family, with no evidence of consanguineous marriage. Thirteen other family members underwent routine examinations, including blood biochemistry, electrocardiograms, carotid and renal artery ultrasounds, echocardiography, and coronary CTA. Blood and biochemical data were analyzed, and a genetic pedigree was established. This study was approved by the Ethics Committee of Fujian Provincial Hospital, and all participating family members provided written informed consent.

### Candidate gene mapping and variant screening strategy

Peripheral blood was collected from the proband in an EDTA anticoagulant tube, and DNA was extracted using the QIAamp DNA Blood Mini Kit (QIAGEN, Cat No. 51106; Qiagen, Hilden Germany) following the manufacturer's protocol. The quality of the extracted DNA was assessed with a NanoDrop spectrophotometer (Thermo Fisher, Waltham, MA, USA). The TargetSeq® liquid phase probe hybridization capture technique developed by iGeneTech® (Beijing, China) was used to build a genomic DNA library capturing the exon regions of 20,000 genes associated with genetic diseases. Sequencing was performed on a NovaSeq 6000 platform with a PE150 sequencing strategy for parallel gene analysis. The gene panel focused on *PCSK9*, *APOB*, *LDLR*, *LDLRAP1*, ATP-binding cassette sub-family G member 5 (*ABCG5*), and *ABCG8*. Following sequencing, data were aligned to the reference Human hg19 genome. Single-nucleotide variants and small indels were identified and analyzed using GATK and ANNOVAR, with further screening and annotation conducted per ACMG guidelines. The variant data were cross-referenced with public databases such as dbSNP, 1000 Genomes (1000G), the Human Gene Mutation Database, and the Exome Sequencing Project v. 6500 (ESP6500). Filtered sequencing reads were aligned to the hg19 reference genome using the Burrows-Wheeler Aligner, and variant information was analyzed in terms of gene features, variant type, and their frequency in 1000G and ESP6500. Predictive tools, including Polymorphism Phenotyping (PolyPhen-2, http://genetics.Bwh.harvard.edu/ppH2/), sorting intolerant from tolerant (SIFT, http://sift.jcvi.org/), Protein Variation Effect Analyzer (PROVEAN, http://provean.jcvi.org/protein_batch_submit.php?species=human), and mutation taster (http://mutationtaster.org/), were used to assess the pathogenicity and biological significance of the identified variants.

### Sanger sequencing

Sanger sequencing was performed on fourteen family members, including the proband, to validate the suspected variant sites. Primers specifically targeting these variants were designed utilizing Primer Premier 5.0 software. To verify the NovaSeq 6000 sequencing results, Sanger sequencing was conducted using the ABI 3500 Dx genetic analyzer. The amplified fragment of the target sequence of the *LDLR* (NM_000527) c.G1027A variant was 322 bp. The forward primer(F) was GAGTGACCAGTCTGCATCCC and the reverse primer(R) was AAACTCTGGCCAGCCAATGA, both with an annealing temperature of 60°C. The amplified target sequence for the LDLR c.G1879A variant measured 1122 bp, with a forward primer (F)

sequence of GAAATGGATGGTGGTGATG and a reverse primer (R) sequence of GGGCAGAA GAAGCGGAGT, also with an annealing temperature of 60˚C. All primers were synthesized by Genokon Medical (Xiamen, China).

## Construction and identification of *LDLR* wild type (WT), p.Gly343Ser and p.Ala627Thr mutant plasmids

The pCDH-CMV-MCS-EF1-copGFP-T2A-puro plasmid synthesis scheme was used as the expression vector to synthesize *LDLR*, with a XbaI/NotI cleavage site. Two plasmids were constructed: WT plasmid 1 (pCDH-CMV-hLDLR-WT-EF1-copGFP-T2A-Puro) and mutant plasmid 2 (pCDH-CMV-hLDLR-G1027A-EF1-copGFP-T2A-Puro/pCDH-CMV-hLDLR-G1879A-EF1-copGFP-T2A-Puro), both containing XbaI/NotI cleavage sites. The mutant plasmid 2 contained the 1027G>A, 1879G>A mutation in the *LDLR* gene. Target genes were amplified and sequenced to confirm the introduction of mutations. The cloning of *LDLR* (WT) and *LDLR* (1027G>A,1879G>A), as well as the synthesis of related polymerase chain reaction (PCR) primers, were performed by Wuhan Gene Create Biological Engineering Co. Ltd. (Wuhan, China).

## Cell transfection

HEK293T cells were digested and collected using trypsin, then seeded into a 10 cm Petri dishes at a density of $6-7 \times 10^6$ cells per dish in an appropriate complete culture medium. After cell attachment, the culture reached 80–90% confluence. Cells were incubated at 37˚C with 5% $CO_2$ overnight. Once the cells adhered to 70–90%, transient transfection was initiated. A mixture of 12 μg lentiviral plasmid and 12 μg Lenti-Mix(where Lenti-Mix consists of pMDLg/pRRE:pVSV-G:pRSV-Rev = 5:3:2 ratio) was prepared in a 5 mL EP tube, gently mixed, and left at room temperature for 10 minutes. Next, TurboFect-DNA Mix was added to the culture dish and the complete medium was replaced after 12 h. Following a 48-hour incubation, the cells were observed under a microscope and the medium was collected for the next step.

## Quantitative real-time PCR

Total RNA was extracted from HEK293T cells using the TriPure Isolation Reagent kit (Roche, Shanghai, China; Catalog No. 11,667,165,001). LDLR transcription levels were assessed via reverse transcription and quantitative real-time PCR (qRT-PCR). The first chain of cDNA was synthesized according to HiFiScript (CW2020M, CWBIO, Beijing, China). The reaction system contained 2.5 mM dNTP Mix, 4 μL; primer mix, 2 μL (primers in Table 1); RNA Template, 7 μL; 5× RT Buffer, 4 μL; 1× dithiothreitol, 0.1 M, 2 μL; 10 mM HiFiScript, 200 U/μL; and RNase-free water, 20 μL. After mixing with a vortexer, the tube was briefly centrifuged. The reaction was incubated at 42˚C for 50 minutes and then at 85˚C for 5 minutes. The resulting cDNA was diluted 20-fold, and 40 RT-qPCR cycles were performed in a Roche LightCycler 480 (Roche, Beijing, China).

**Table 1. Primers for quantitative real-time PCR.**

| Primers for quantitative real-time PCR | primer sequence |
| --- | --- |
| hLDLR qRT F | 5`- AAGTGCATCTCTCGGCAGTT -3` |
| hLDLR qRT R | 5`- CCCCTTGGAACACGTAAAGA -3` |
| hGAPDH F | 5`- CAAGGTCATCCATGACAACTTTG -3` |
| hGAPDH R | 5`- GTCCACCACCCTGTTGCTGTAG -3` |

## Western blot

HEK293T cells were cultured and lysed to extract total protein, which was used to assess LDLR expression. Protein samples were separated by electrophoresis and transferred to a polyvinylidene fluoride (PVDF) membrane, The membrane was then blocked at room temperature for 1 hour using 5% skim milk in Tris-buffered saline with 0.1% Tween® 20 (TBST). Next, the membrane was washed once and incubated with primary antibodies: Anti-a-Tubulin antibody (ab7291, Abcam, UK,1:3000) or actin antibody (ab8227, Abcam, UK, 1: 5000). Flag antibody (F1804, SIGMA, USA, 1:1000 diluted with 5% bovine serum albumin (BSA)), was added and incubated overnight at 4°C, and the membrane was then washed three times. Horseradish peroxidase-labeled secondary antibodies (Goat anti Rabbit IgG 1:2000, Goat anti Mouse IgG 1:2000, diluted with 5% BSA, ab6721, ab6789, Abcam, UK), diluted with 5% BSA, were applied and incubated with gentle shaking at room temperature for 1 hour. The PVDF membrane was washed five times with TBST and once with deionized water (ddH2O) before detection.

## Enzyme linked immunosorbent assay (ELISA) of LDLR in HEK293T cell lysates and cell supernatants

LDLR expression in the conditioned medium of HEK293T cells was quantified by a commercial ELISA following the indications of the manufacturer (ab190808, Abcam, UK). The parameters of the microplate reader (Varioskan Lux, Thermo, Massachusetts, USA) configured as recommended. Optical density at 450 nm was measured immediately after the reaction was completed. A standard curve was generated, and LDLR levels in the samples were calculated based on this curve.

## Immunofluorescence localization assay

After fixation, permeabilization, and blocking, the transfected HEK293T cells were incubated overnight at 4°C with an Anti-FLAG antibody (diluted 1:1000). The cells were then rinsed three times with phosphate-buffered saline (PBS) before being incubated with a fluorescent secondary antibody (diluted 1:1000) at room temperature in the dark for 2 hours. Following three additional washes with PBS, the cells were stained with 4′,6-diamidino-2-phenylindole (DAPI). The staining was carried out at room temperature for 5 minutes, with two rinses using 1× PBS for 3 minutes each. Images were captured using a laser confocal microscope (Nikon A1, Shanghai, China).

## Statistics

Experimental data were statistically analyzed using GraphPad Prism 6.02. An unpaired t-test was used to compare the two groups. The mean value was expressed as the mean ± standard error of the mean (SEM), and $p < 0.05$ indicated that the difference was statistically significant.

## Results

### Pedigree clinical data

There were 14 members in the family, including five males and nine females, and eight members, including the proband, diagnosed with hypercholesterolemia. The proband presented with a 20-year history of recurrent chest tension and had a soybean-sized xanthoma on the left upper eyelid with a corneal arcus (Fig 1A–1c). After admission, routine blood tests, liver function, renal function, electrolytes, and blood glucose tests were within normal ranges.

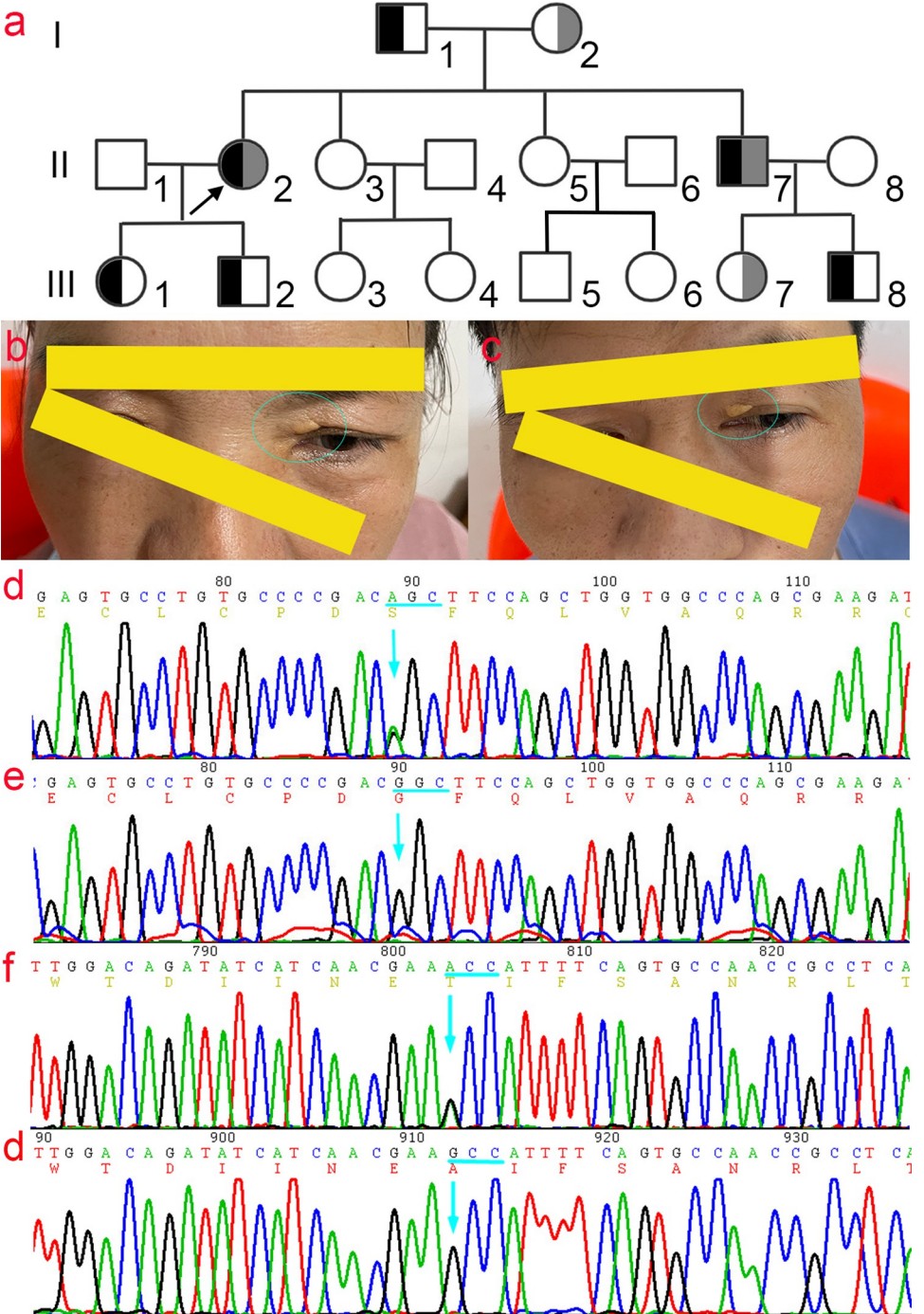

**Fig 1. Family genogram of FH and the Sanger sequencing map.** a) Family genogram of FH. The proband (II2) and younger brother (II7) carrying compound heterozygotes; black and grey indicate c.G1027A (p.G343S) and c.G1879A (p.A627T) mutation in *LDLR* gene, respectively. b) and c) The symptoms of xanthomas on the eyelid skin of the proband (b) and brother (c). d) The Sanger sequencing map of the c.G1027A (p.G343S) mutant and e) The corresponding wild-type Sanger sequencing map. f) The Sanger sequencing map of the c.G1879A (p.A627T) mutant and g) The corresponding wild-type Sanger sequencing map.

Autoantibodies and anti-O antibodies were negative. An immune panel, erythrocyte sedimentation rate, and C-reactive protein levels were also normal. Thyroid function tests, including thyroid-stimulating hormone, triiodothyronine, and thyroxine, were unremarkable, and the electrocardiogram showed no obvious abnormalities. Blood lipid profiles revealed: total cholesterol(TC) 20.50 mmol/L ($<$ 5.18 mmol/L), triglyceride (TG) 1.45 mmol/L ($<$ 1.7 mmol/L), LDL-C 18.80 mmol/L, high-density lipoprotein cholesterol (HDL-C) 1.26 mmol/L (1.29–1.55 mmol/L), apolipoprotein A (apoA) 1.53 g/L (1.2–1.6 g/L), apolipoprotein B (apoB) 0.97 g/L (0.6–1.1 g/L). Coronary computed tomography angiography (CTA) revealed segmental eccentric mixed plaques in the wall of the proximal segment of the left main coronary artery to the anterior descending branch, with stenosis of approximately 70%–80%. Echocardiography revealed left ventricular enlargement, moderate mitral insufficiency, and decreased coronary flow reserve. Carotid artery color Doppler ultrasound indicated plaques at the origin of the bilateral common carotid artery and diffuse thickening of the intima in the middle artery. Renal artery ultrasound detected scattered plaques at the renal artery ostium and along the main trunk, with color Doppler flow imaging (CDFI) showing adequate blood flow within these areas. Based on the British Simon Broome criteria [14] and the Dutch Lipid Clinic Network criteria [15] for FH, combined with family history and gene sequencing results, the proband was diagnosed with FH and coronary atherosclerotic heart disease, exhibiting a compound heterozygous mutation.

The younger brother of the proband presented with cutaneous xanthomas and coronary atherosclerotic heart disease. Eight individuals, including the proband (II2), were diagnosed with hypercholesterolemia (I1, I2, II2, II7, III1, III2, III7, and III8). Carotid color ultrasound of the proband's father (I1) revealed plaques at the bifurcation of the left common carotid artery. Doppler ultrasound of the carotid artery of the proband's younger brother (II7) revealed bilateral common carotid artery plaques and right subclavian artery plaques. Echocardiography revealed mild mitral and tricuspid regurgitation, while renal artery ultrasound showed scattered plaques along the arterial wall. Coronary CTA showed segmental eccentric mixed plaques in the proximal segment of the left main coronary artery extending to the anterior descending branch, with an estimated stenosis of 65%–75%. No significant abnormalities were detected in other family members. The plasma cholesterol levels in four individuals met the diagnostic criteria for FH. Among these individuals, both the proband and her younger brother exhibited TC and LDL-C levels indicative of homozygous FH (HoFH), while the father and son presented with values consistent with heterozygous FH (HeFH). Serum total cholesterol levels were 20.50–21.35 mmol/L in family members with compound heterozygous mutations (II2, II7), 6.52–8.81 mmol/L in those with a single heterozygous mutation (I1, I2, III1, III2, III7, III8), and 2.94–3.61 mmol/L in mutation-free family members(II3-6, II8, III3-6) (Table 2).

## High-throughput genome sequencing and verification using Sanger sequencing

Target enrichment with next-generation sequencing(NGS) was used to screen for pathogenic gene variants and copy number variations (CNVs) across all exons of the genome. To identify pathogenic variants in the proband's genome, single nucleotide polymorphisms (SNPs) with a minor allele frequency greater than 1% were excluded. A heterozygous missense variant, c.1027G$>$A, was identified in exon 7 of the *LDLR* gene(NM_000527), and another heterozygous missense variant, c.1879G$>$A, was identified in exon 13. Both variants were confirmed by Sanger sequencing (Fig 1D–1G). The c.1027G$>$A and c.1879G$>$A variants result in the substitution of glycine (Gly) with serine (Ser) at residue 343 and alanine (Ala) with threonine (Thr) at residue 627 of the LDLR protein, respectively. These variants have been both included

**Table 2. The results of the blood lipid test of family members.**

| Family number | Age | Variants | TC (mmol/l) | TG (mmol/l) | HDL-C (mmol/l) | LDL-C (mmol/l) | ApoA (g/l) | ApoB (g/l) |
|---|---|---|---|---|---|---|---|---|
| I1 | 77 | p.Gly343Ser | 8.75 | 0.67 | 1.28 | 7.27 | 1.34 | 0.45 |
| I2 | 75 | p.Ala627Thr | 6.81 | 2.12 | 0.93 | 5.31 | 1.16 | 1.32 |
| II2 | 56 | p.Gly343Ser, p.Ala627Thr | 20.50 | 1.45 | 1.26 | 18.80 | 1.53 | 0.97 |
| II3 | 53 | - | 2.94 | 0.76 | 1.35 | 1.42 | 1.51 | 0.40 |
| II5 | 50 | - | 3.61 | 0.46 | 2.11 | 1.37 | 1.76 | 0.44 |
| II7 | 48 | p.Gly343Ser, p.Ala627Thr | 21.35 | 2.27 | 0.58 | 20.29 | 1.20 | 1.45 |
| III1 | 25 | p.Gly343Ser | 6.62 | 1.22 | 1.12 | 4.92 | 1.10 | 1.21 |
| III2 | 23 | p.Gly343Ser | 8.81 | 1.02 | 1.10 | 7.48 | 1.34 | 1.17 |
| III3 | 28 | - | 3.60 | 1.53 | 0.63 | 2.49 | 1.16 | 0.93 |
| III4 | 21 | - | 3.27 | 0.93 | 1.10 | 2.04 | 1.31 | 0.62 |
| III5 | 23 | - | 3.18 | 1.16 | 1.03 | 1.94 | 1.19 | 0.73 |
| III6 | 22 | - | 3.06 | 1.38 | 0.76 | 2.02 | 0.93 | 0.70 |
| III7 | 22 | p.Ala627Thr | 7.58 | 1.38 | 1.08 | 6.11 | 1.24 | 0.82 |
| III8 | 20 | p.Gly343Ser | 6.97 | 0.87 | 1.68 | 5.03 | 1.61 | 0.54 |
| Normal reference value | | | <5.18 | <1.7 | 1.04–1.55 | 0–3.37 | 1.2–1.6 | 0.6–1.1 |

Note

[a]TC, Total cholesterol.

[b]TG, Triglycerides.

[c]HDL-C, High density liptein cholesterol.

[d]LDL-C, Low Density Lipoprotein.

[e]apoA, Apolipoprotein B.

[f]apoB, Apolipoprotein B.

in the ClinVar database. According to the ClinGen FH Variant Curation Expert Panel (VCEP) criteria [16], the c.1027G>A variant is classified as Pathogenic, while the c.1879G>A variant is classified as Likely Pathogenic. Sanger sequencing of family members showed that the proband and her brother (II7) carried compound heterozygous missense mutations: c.1027G > A (p.Gly343Ser) and c.1879G > A (p.Ala627Thr), whereas I1, III1, III2, and III8 only carried the c. G1027A mutation, and I2 and III7 carried the c. G1879A mutation. No mutations were identified in other family members (Fig 1A).

## Cloning of LDLR wild-type and p.Gly343Ser, p.Ala627Thr gene mutants

The cloning and eukaryotic expression vectors of LDLR-WT, LDLR (p.Gly343Ser, p. Ala627Thr) were successfully constructed. The fragments generated by XbaI/NotI digestion of LDLR-WT, mutant LDLR (p.Gly343Ser, p.Ala627Thr) were approximately 2666 bp, which was consistent with the design. The constructed vector was verified by sequencing and was transfected into HEK293T cells (Fig 2B–2D).

## Detection of transfection efficiency of pCDH lentiviral expression vector

HEK293T cells in the logarithmic growth phase were infected with various lentiviral expression vectors, including pCDH lentiviral empty load, pCDH-CMV-LDLR (WT)-EF1-copGFP-T2A-Puro, pCDH-CMV-LDLR (c.G1027A)-EF1-copGFP-T2A-Puro, and pCDH-CMV-LDLR (c.G1879A)-EF1-copGFP-T2A-Puro. After 48–72 hours of infection, high levels of green fluorescent protein (GFP) expression were observed in the cells, indicating successful transduction and robust cell growth (Fig 2A).

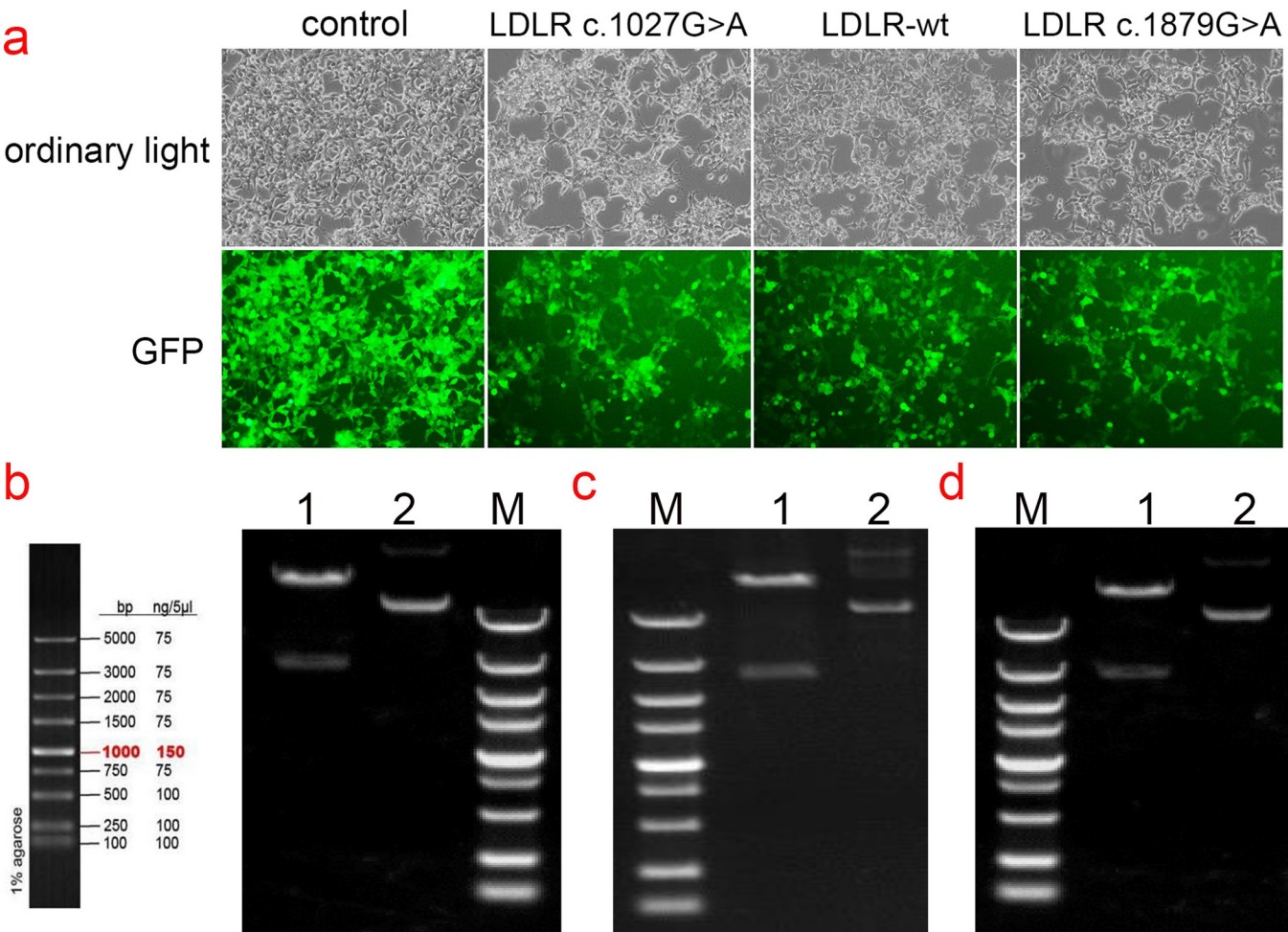

**Fig 2. GFP expression in HEK293T cells transfected with plasmids and Electropherogram of WT and mutant plasmids digested with XbaI/NotI.** (a) Green fluorescent protein expression after transfection of cells carrying unloaded (Control), wild-type (LDLR -WT), Gly343Ser mutant (LDLR- c.1027G>A), and Ala627Thr mutant (LDLR- 1879G>A), suggesting that there was no significant difference in transfection efficiency among the groups. (b) Electropherogram of wild-type plasmid digested with XbaI/NotI; (c)Electropherogram of mutant c.1027G>A plasmid digested with XbaI/NotI; (d) Electropherogram of mutant c. 1879G>A plasmid digested with XbaI/NotI, Lane 1: undigested electropherogram, Lane 2: electropherogram after double digestion, Lane 3: DNA marker.

## Intracellular localization of LDLR wild-type and mutants

The intracellular localization of WT and mutant LDLR was detected by immunofluorescence (Fig 3). The results demonstrated that LDLR was predominantly distributed in the cytoplasm of the cells. Notably, the p.Ala627Thr mutant displayed significantly reduced fluorescence intensity compared to the wild-type, suggesting decreased protein expression. In contrast, the p.Gly343Ser mutant did not show a significant difference in intracellular fluorescence localization relative to the WT.

## Expression of WT and mutant LDLR in HEK293T cells

The relative mRNA expression of the WT (LDLR-wt) and p.Gly343Ser, p.Ala627Thr mutant (LDLR-mut) LDLR in HEK293T cells was measured using qRT-PCR. The results showed that LDLR mRNA expression was downregulated in cells carrying the Gly343Ser and Ala627Thr mutations compared to wild-type cells ($P < 0.05$) (Fig 4A). Western blotting was performed to

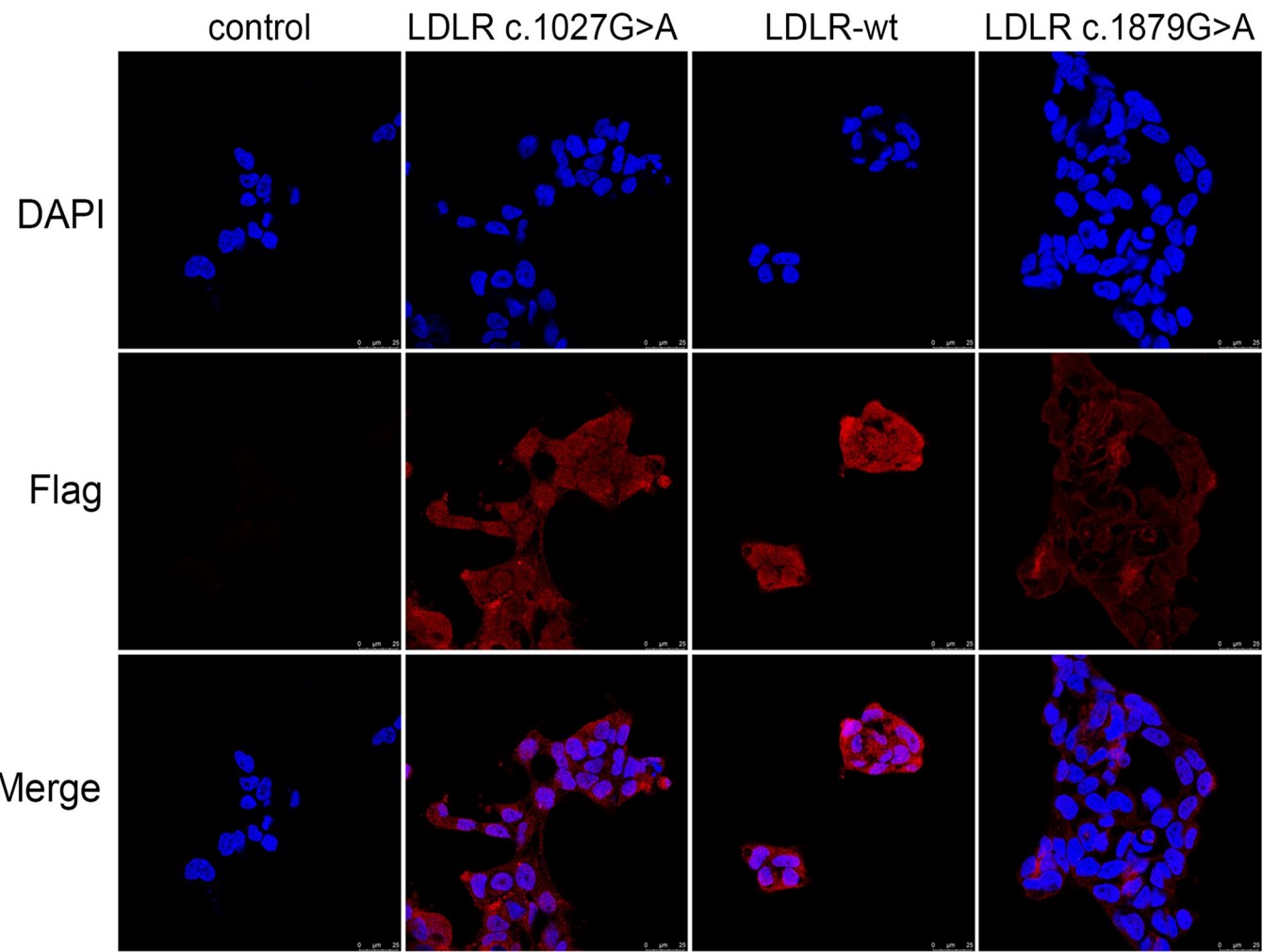

**Fig 3. Immunofluorescence analysis of LDLR protein localization in HEK293T cells.** Immunofluorescence was used to detect the localization of LDLR protein in HEK293T cells carrying unloaded (Control), wild-type (LDLR-WT), Gly343Ser mutant (LDLR- c.1027G>A), and Ala627Thr mutant (LDLR-1879G>A). The p.Ala627Thr mutants were significantly attenuated compared to that of the wild-type, suggesting that protein expression was similarly attenuated; There was no significant difference in intracellular fluorescence localization before and after introduction of the p.Gly343Ser mutation and the proteins were localized in the cytoplasm.

assess LDLR expression in both cell lysates and culture supernatants, which revealed a substantial reduction in LDLR expression associated with the mutations (Fig 4D). Additionally, the expression of LDLR in the cell culture medium supernatant and cell lysate was detected by ELISA (Fig 4B and 4C). The results showed no significant difference in LDLR protein levels in the culture supernatant between the mutant and wild-type groups. However, LDLR protein levels in the cell lysate of the Gly343Ser and Ala627Thr variants groups were significantly lower than those in the wild-type group, which is consistent with the western blotting findings.

## Structure and function prediction of the mutant LDLR

The interaction diagram of LDLR with other proteins was plotted using String(https://string-db.org/cgi/network?taskId=bniuzKD4qlSX&sessionId=b8YfT7vyWuWv) and the X-ray crystal diffraction structure of the LDLR in complex with proprotein convertase subtilisin/kexin type 9 (PCSK9) was plotted using Swiss-Model(https://swissmodel.expasy.org/repository/uniprot/P01130) (Fig 5A and 5B). The location of the mutation site on the three-dimensional

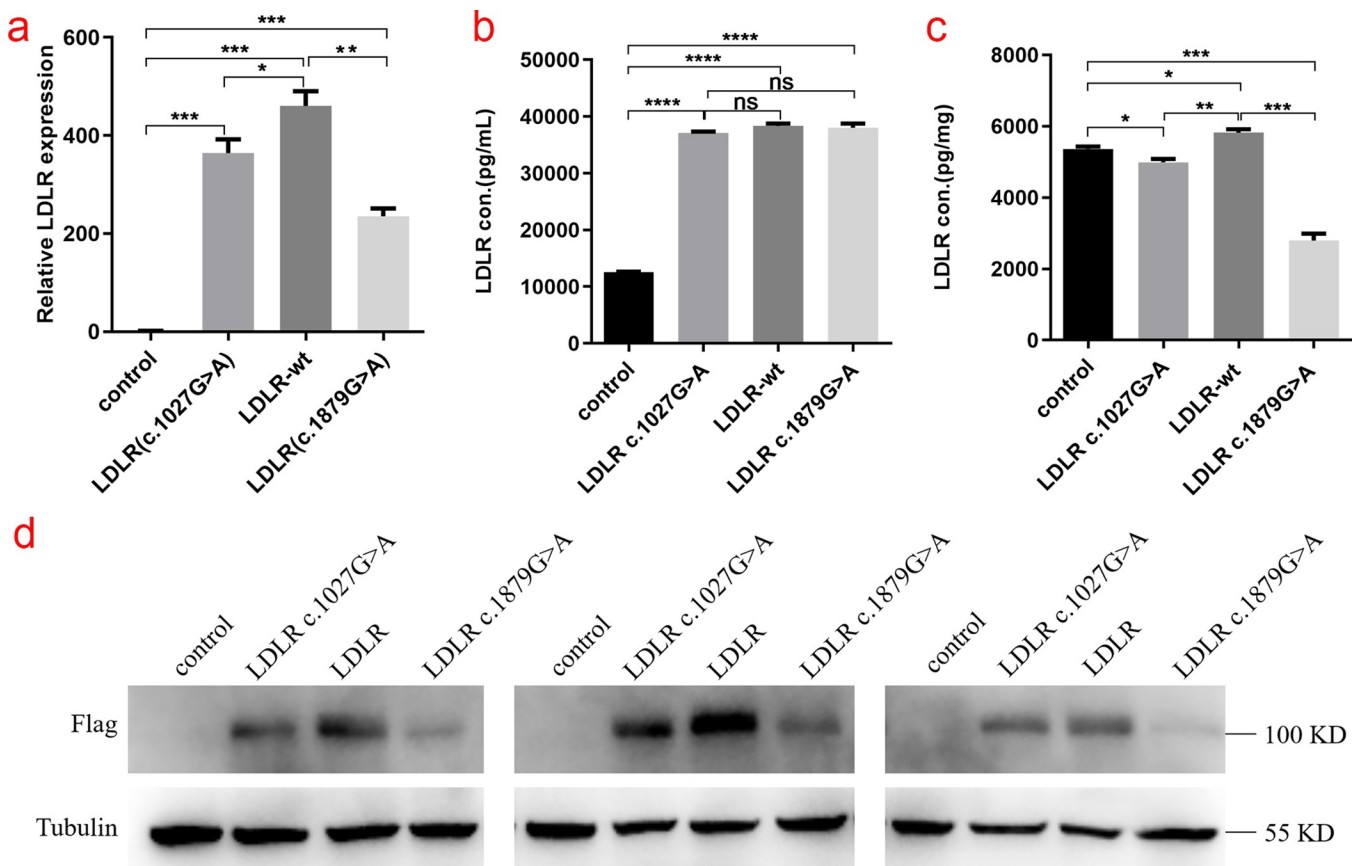

**Fig 4. Relative LDLR mRNA and protein expression of control, LDLR-WT, Gly343Ser mutant and Ala627Thr mutant in HEK293T cells.** (a) Relative LDLR mRNA expression levels (*, P < 0.05, **, P < 0.01, ***, P < 0.001) in HEK293T cells by Control, LDLR-WT, Gly343Ser mutant (LDLR- c.1027G>A), and Ala627Thr mutant (LDLR- 1879G>A). (b, c) Based on the enzyme-linked immunosorbent assay (ELISA) standard curve, the expression of Control, LDLR-WT, Gly343Ser mutant (LDLR- c.1027G>A), and Ala627Thr mutant (LDLR- 1879G>A) in the supernatant of HEK293T cells was calculated and cell lysate. (*, P < 0.05, **, P < 0.01, ***, P < 0.001) (d) Western blotting was used to detect the expression of LDLR protein in HEK293T cells by Control, LDLR-WT, Gly343Ser mutant (LDLR- c.1027G>A), and Ala627Thr mutant (LDLR- 1879G>A), respectively.

structure of the LDLR is depicted in Fig 5B. The protein structure near the two mutation sites was simulated using Chimera(https://community.chocolatey.org/packages/chimera/1.15), which showed that these two mutations had a minimal impact on LDLR protein structure (Fig 5C and 5D). In addition, the effect of the two mutation sites on the functional domain of the protein was predicted using Predictprotein software(https://predictprotein.org/). The G343S and A627T mutations were found to affect the disordered region, protein interaction binding region, and RNA binding region of the LDLR, suggesting that these mutations might influence the properties and function of the protein (S1 Fig).

Protein sequence conservation analysis showed that the mutation sites Gly343 and Ala627 were highly conserved in mammals(https://www.uniprot.org/uniprot/P01130). Mutations near Gly343 would cause severe LDLR dysfunction, indicating that these two amino acid sites played a crucial role in maintaining LDLR function (Fig 6A). Phosphorylation modification of these two sites was predicted using GPS 5.0(http://gps.biocuckoo.cn/). G343S was recognized and phosphorylated by protein kinase signaling pathways, such as IκB kinase and cAMP-dependent protein kinase A/cGMP-dependent protein kinase G/protein kinase C (Fig 6B), while A627T was recognized and phosphorylated by protein kinase signaling pathways, such as sterile and tyrosine kinase-like (Fig 6C). In addition, A627T might affect phosphorylation at

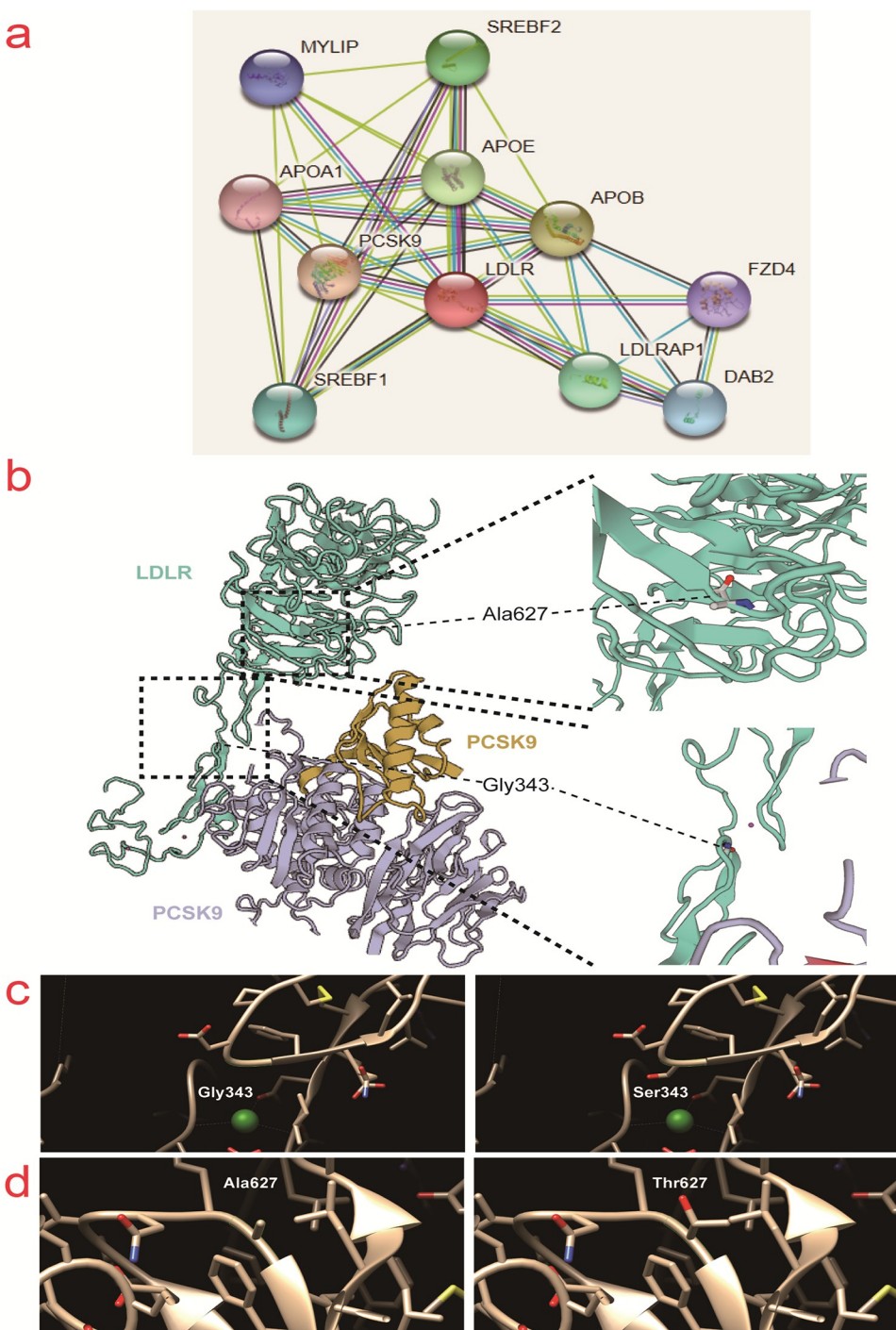

**Fig 5. Effects of LDLR Gly343Ser and Ala627Thr mutations on the protein tertiary structure.** a) The LDLR interacting proteome included in the String database (https://string-db.org/cgi/network?taskId=bniuzKD4qlSX&sessionId=b8YfT7vyWuWv). b) The tertiary structure of LDLR (green) and its interacting protein PCSK9 (yellow and purple) was obtained from Swiss-model database (https://swissmodel.expasy.org/repository/uniprot/P01130). c) Chimera 1.15 (https://community.chocolatey.org/packages/chimera/1.15) predicted changes in LDLR tertiary structure by the mutation of Gly343 to Ser343. d) Chimera 1.15 predicted changes in LDLR tertiary structure by the mutation of Ala627 to Thr627.

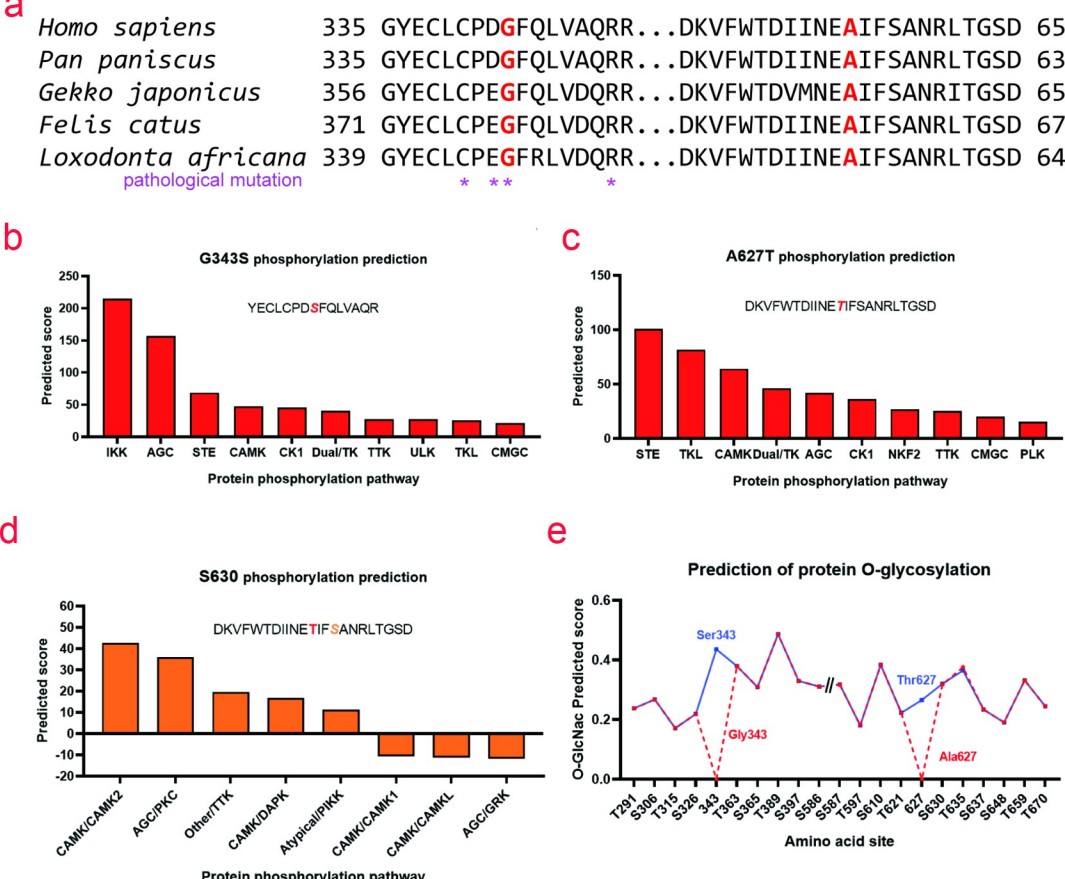

**Fig 6. Conservation analysis of sequences near the mutation sites and effects of LDLR Gly343Ser and Ala627Thr mutations on protein post-translational modification.** a) Conservation analysis of sequences near the mutation sites of Gly343Ser and Ala627Thr. * indicates pathogenic mutation loci included in the Uniprot database (https://www.uniprot.org/uniprot/P01130). b) GPS 5.0 (http://gps.biocuckoo.cn/) predicted signaling pathway regulating phosphorylation of Gly343Ser mutation site. c) GPS 5.0 predicted the phosphorylation signaling pathway function on the Ala627Thr mutation site. d) GPS 5.0 predicted the effect of Ala627Thr mutation onphosphorylation signaling pathway of the Ser630 site. e) YinOYang-1.2 (https://services.healthtech.dtu. dk/service.php?YinOYang-1.2) predicted the effect of LDLR Gly343Ser and Ala627Thr mutations on protein O-glycosylation.

S630 (Fig 6D). YinOYang 1.2(https://services.healthtech.dtu.dk/service.php?YinOYang-1.2) was used to predict that these two sites might undergo O-GlcNac modification (Fig 6E). The potential changes in protein post-translational modification of the mutation sites and adjacent amino acids could alter the structure, function, localization, and fate of the proteins, leading to functional defects in LDLR.

## Discussion

Familial hypercholesterolemia (FH) represents the most prevalent congenital lipid metabolism disorder, predisposing individuals to premature atherosclerosis and coronary artery disease (CAD), frequently resulting in early mortality [17]. The intensity of hypercholesterolemia in patients with FH is related both to the function of the mutant protein, and the quantity of defective alleles [18]. Nonsense mutations in the *LDLR* gene are associated with the most severe cases, whereas pathogenic variants in the *APOB and PCSK9* genes typically result in milder phenotypes [19]. Individuals harboring biallelic pathogenic variants in the *LDLR* gene typically suffer from a more severe form of the disease, whereas those with a single

heterozygous pathogenic variant generally display milder symptoms. More severe phenotypes can also arise from the interaction of pathogenic genes, with the severity of FH varying based on the inheritance pattern and gene dosage effects associated with the *LDLR* gene [20]. In this context, variants are classified as semi-dominant pathogenic for FH.

The *LDLR* gene, first cloned in 1983, is located on chromosome 19 at positions 19p13.2–13.3, spanning 45 kilobases and comprising 18 exons and 17 introns [21,22]. It encodes a receptor that is essential for the clearance of LDL-C from the blood plasma. Over 2200 mutations in the *LDLR* gene have been identified, which include point mutations, deletions, and complex rearrangements (www.ucl.ac.uk/fh). Mutations predominantly occurred in exon 4, the longest exon in the *LDLR* gene. This critical region encodes the ligand-binding domain, where mutations significantly compromise the functional integrity of the gene [23]. These mutations lead to defective synthesis, maturation, or functioning of the LDL receptor, each contributing differently to the phenotypic spectrum of FH.

Previous research has demonstrated variable impacts of different *LDLR* mutations on receptor function, generally correlating with the nature and position of the mutation in the gene [24]. Strøm [25] revealed that the p.L799R mutation may disrupt the ability of the Sec61 translocon complex to accurately recognize and integrate the mutant LDLR into the membrane, leading to its extracellular secretion. Mutations like p.I792F, p.P795H, and p.V797L exhibit minimal impact on LDLR function, whereas others including p.803_808dup, p.L804P, and p.V806D significantly influence γ-secretase cleavage, a critical process for LDLR functionality [25]. Located within the epidermal growth factor-like domain, the p. Asp360His mutation potentially hindered the internalization and recycling of the LDLc-LDLR complex [26]. The c.226 G>C mutation had no significant impact on LDLR expression or LDL internalization. However, it impaired the receptor's ability to uptake LDL. Conversely, the c.1003 G>T mutation may affect LDLR expression and further weaken its LDL uptake capacity [27]. The c.2389G>A mutation in exon 16 resulted in mRNA miscleavage, causing the mutant LDLR protein to be retained in the Golgi apparatus, disrupting normal cellular processing and receptor functionality [28]. The p.W483X mutation resulted in a truncated LDLR protein that lacked critical domains necessary for normal function, disrupting cholesterol metabolism [29]. However, specific mutations in the *LDLR* gene, such as the 2.5-kb deletion in its 3' untranslated region (UTR), have been shown to exert beneficial effects on gene expression, positively regulating cholesterol metabolism and reducing cardiovascular risk. This deletion modifies the polyadenylation site of *LDLR* mRNA, leading to the loss of miRNA binding sites that have negative effects on gene expression. Consequently, this alteration increases mRNA stability and translational efficacy, which in turn enhances the surface expression of the LDL receptor in cellular membranes [30,31].

The two missense mutations p.Gly403Ser and p.Ala627Thr were found in this study, and their cellular activity, localization, and secretion were examined. The mRNA expression and protein expression of the p.Gly343Ser and p.Ala627Thr mutants were significantly reduced, particularly the p.Ala627Thr mutant, which results in impaired LDLR protein function, according to the results of qRT-PCR and Western blotting.

The ELISA results demonstrated that there was no significant difference in LDLR protein expression in the supernatant of cell culture medium between the mutant groups and wild-type group, indicating the secretion pathway might remain unaffected. However, the expression level of LDLR protein in the cell lysate in the Gly343Ser and Ala627Thr variant groups was prominently lower than that in the wild-type group. Thus, we speculated that p.Gly343Ser and p.Ala627Thr mutant proteins were degraded intracellularly. This observation is critical as it delineates the degradation process as primarily intracellular and not due to secretion inefficiencies. The p.Ala627Thr mutation is more likely to affect LDLR protein expression and

function than the p.Gly343Ser mutation, according to the results of the Western blot and ELISA, which show significantly lower protein expression of the p.Ala627Thr mutant in comparison to the p.Gly343Ser mutant.

Immunofluorescence revealed that the intracellular levels of the p.Ala627Thr mutant were observably elevated compared to the wild-type, whereas the p.Gly343Ser mutation did not manifest a substantial difference in protein expression relative to the wild-type group. The results suggested an increased intracellular localization of the p.Ala627Thr mutant, in spite of a remarkable reduction in protein levels. This indicates a tendency for the p.Ala627Thr mutant to mislocalize or aggregate intracellularly, possibly disrupting cellular processing and trafficking pathways.

The findings of remarkably decreased mRNA and protein levels were inconsistent with studies, such as those by Soutar(18), which highlighted mRNA destabilization and increased protein turnover as common consequences of deleterious *LDLR* mutations. The pronounced decrease in protein levels, especially for p.Ala627Thr, indicated that mutation-induced misfolding or conformational anomalies presumably predispose the LDLR to recognition and degradation by cellular quality control mechanisms. The potential competition between mutant and wild-type LDLR proteins for LDL binding could exacerbate functional deficits. This dominant-negative effect where the mutant protein interferes with the normal function of the wild-type protein could significantly contribute to the phenotypic severity observed in patients with these mutations. The observed intracellular degradation coupled with potential dominant-negative effects substantially increases the complexity of the pathogenic mechanisms and underscores the mutation-specific impacts on the cellular processing of the LDLR.

Our study corroborates previous researches regarding the pathogenic effects of the p.Ala627Thr and p.Gly343Ser mutations in the *LDLR* gene, which are implicated in familial hypercholesterolemia. This research aligns with prior investigations that characterized the impact of the mutations on lowering LDL receptor activity, thereby increasing plasma LDL levels and promoting atherosclerosis [32,33]. Cheng [32] elucidated the specific molecular mechanisms impacted by the p.Ala627Thr mutation, suggesting that this variant may contribute to FH by promoting ROS/NLRP3-mediated pyroptosis in hepatic cells. Empirical studies have demonstrated that the accumulation of intracellular cholesterol crystals dramatically amplifies reactive oxygen species (ROS) levels, which subsequently trigger the activation of the NLRP3 inflammasome, a key component in the inflammatory response [34]. This activation specifically induces pyroptosis in hepatic cells, leading to widespread inflammatory responses at both the cellular and tissue levels, characterized by cytokine release and immune cell recruitment [32,35,36]. These pathological processes result in hepatic tissue damage and critically exacerbate the clinical manifestations of familial hypercholesterolemia, including accelerated atherosclerosis and increased cardiovascular risk.

Despite its high prevalence, FH remains severely neglected and undertreated. Estimates suggest that more than 90% of the approximately 40 million individuals globally affected by FH remain undiagnosed [19]. Consequently, it is imperative to disseminate knowledge about FH widely and actively enhance diagnostic rates. In this context, employing cascading genetic screening, an effective and cost-efficient strategy, plays a pivotal role in identifying individuals with FH and profoundly reducing the risks of coronary heart disease, myocardial infarction, and premature mortality [37–39]. Moreover, in-depth investigation of *LDLR* gene mutations and their pathogenic mechanisms is essential, as it not only sharpens clinicians' understanding and improves treatment strategies for FH but also significantly contributes to the advancement of disease databases and the elucidation of intricate molecular pathogenic mechanisms.

Several limitations merit discussion. First, the research requires detailed in vitro molecular analyses to elucidate the specific impacts on transcription and splicing processes, and

comprehensively assess the efficacy of pharmacological treatments for FH. Secondly, while the functional assays employed provide valuable insights, they fall short of mimicking the intricate, multi-regulatory interactions characteristic of the in vivo environment. Furthermore, the limited sample size constrains the generalizability of our findings. Expanding the cohort of FH patients is required to fully explore the diverse spectrum of pathogenic variants among FH patients in China.

## Conclusions

In this study, we identified the LDLR c.G1027A (p.Gly343Ser), c.G1879A (p.Ala627Thr) heterozygous mutation as pathogenic variants site. These mutations may compromise protein synthesis and accelerate the intracellular breakdown or mislocalization of LDLR, ultimately resulting in diminished LDLR levels and functionality, which contributes to the development of FH.

## Supporting information

**S1 Fig. Effects of mutations on protein functional domains.** Functional domain prediction for wild-type (top) and mutant (bottom) using Predictprotein (https://predictprotein.org/). The shaded area represents the location of the mutation; the red arrows indicate regions where wild-type and mutants vary significantly in functional domains.
(TIF)

## Author Contributions

**Conceptualization:** Li Zhang, Xin-fu Lin.

**Data curation:** Jing Zou, Zhu-ting Fang.

**Formal analysis:** Jian-Hui Zhang.

**Funding acquisition:** Ying Chen, Zhu-ting Fang, Fan Lin, Hong Li.

**Investigation:** Li-Sheng Liao, Hong Li.

**Methodology:** Dan-dan Ruan, Yan-ping Zhang.

**Project administration:** Li-Sheng Liao.

**Supervision:** Li Zhang, Xin-fu Lin, Jie-Wei Luo.

**Writing – original draft:** Ya-nan Hu, Min Wu, Hong-ping Yu, Qiu-yan Wu.

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
