## [Decision Letter · Decision Letter 0]

5 Jul 2024

PONE-D-24-12858Analysis of low-density lipoprotein receptor gene mutations in a family with familial hypercholesterolemiaPLOS ONE

Dear Dr. Luo,

Thank you for submitting your manuscript to PLOS ONE. After careful consideration, we feel that it has merit but does not fully meet PLOS ONE’s publication criteria as it currently stands. Therefore, we invite you to submit a revised version of the manuscript that addresses the points raised during the review process. />

We look forward to receiving your revised manuscript.

Kind regards,

Nejat Mahdieh

Academic Editor

PLOS ONE

Journal Requirements:

2. Thank you for submitting the above manuscript to PLOS ONE. During our internal evaluation of the manuscript, we found significant text overlap between your submission and previous work in the [introduction, conclusion, etc.].

Please revise the manuscript to rephrase the duplicated text, cite your sources, and provide details as to how the current manuscript advances on previous work. Please note that further consideration is dependent on the submission of a manuscript that addresses these concerns about the overlap in text with published work.

[If the overlap is with the authors’ own works: Moreover, upon submission, authors must confirm that the manuscript, or any related manuscript, is not currently under consideration or accepted elsewhere. If related work has been submitted to PLOS ONE or elsewhere, authors must include a copy with the submitted article. Reviewers will be asked to comment on the overlap between related submissions (http://journals.plos.org/plosone/s/submission-guidelines#loc-related-manuscripts).]

We will carefully review your manuscript upon resubmission and further consideration of the manuscript is dependent on the text overlap being addressed in full. Please ensure that your revision is thorough as failure to address the concerns to our satisfaction may result in your submission not being considered further.

6. Please include complete captions for your Supporting Information files at the end of your manuscript, and update any in-text citations to match accordingly. Please see our Supporting Information guidelines for more information: http://journals.plos.org/plosone/s/supporting-information. 

Reviewers' comments:

Reviewer's Responses to Questions

**Comments to the Author**

1. Is the manuscript technically sound, and do the data support the conclusions?

Reviewer #1: Partly

Reviewer #2: Yes

2. Has the statistical analysis been performed appropriately and rigorously? 

Reviewer #1: Yes

Reviewer #2: Yes

3. Have the authors made all data underlying the findings in their manuscript fully available?

Reviewer #1: Yes

Reviewer #2: Yes

4. Is the manuscript presented in an intelligible fashion and written in standard English?

Reviewer #1: Yes

Reviewer #2: Yes

5. Review Comments to the Author

Reviewer #1: The authors reported variants in LDLR gene in a Chinese family with FH. In silico analysis and functional assays were perform. Some major issues need to be addressed.

Row 27 (and in whole body of manuscript): LDLR - gene (genes) name should be writen in italics.

Row 32: Rather „genotype-phenotype associations“ than „genetic-phenotypic associations“.

Rows 49-51: „Conclusions“: Presented information are well known for many years and cannot be presented as conclusions of this study. Please, completely revise the text.

Row 69: Rather “caused by pathogenic variations” than “caused by variations”.

Row 77: Rather “common population” than “average people”.

Row 86: The author is writing about the apolipoprotein APOB, not the gene, so italics cannot be used.

Row 93: 1951? Joseph Leonard Goldstein (* 18. April 1940) and Michael Stuart Brown (* 13. April 1941), so at the age of 11 and 10? – reference [17] is from 1974.

Row 176: Use italics for LDLR.

Row 176: Which “Target genes…”.

Row 218: …of PROS1? Why PROS1? This is the reason: PMID: 34496879? In my opinion, you overwrote (copy/pasted) your older text.

Row 264: “Simon criteria” are “Simon Broome criteria”, “Dutch Network Diagnostic Criteria” are “Dutch Lipid Clinic Network criteria”.

Row 301: “According to the… (ACMG)” – there are VCEP criteria for LDLR published in 2022 (PMID: 34906454).

Row 337: Missing space between “Thr” and “mutant”.

Rows 384 – 566: “Discussion”. This is NOT discussion! Presented information are rather encyclopedic information about LDLR, LDLR protein, LDLR cycle and pathogenic variants in LDLR than discussed results of this study. Why did not you discuss results of assays for Gly343Ser published here PMID: 25647241 or here PMID: 37397863 for Ala627Thr? Please, COMPLETELY revise the text.

Row 520: p.Arg53fs? Why Arg53fs? This is the reason: PMID: 34786791? In my opinion, you overwrote (copy/pasted) your older text.

Row 529: We → we

Row 548: Severity → severity

Row 714: Check the reference [39].

Row 822: Table 1, Head of second column: ago → Age

Comment: It is necessary to mention that the authors are minimally oriented in the field of LDLR or FH. Errors on line 93 or 264 are clear evidence of this. It is necessary to completely revise the body of the manuscript. Please clearly explain how the errors on Rows 218 and 520 occurred.

Reviewer #2: The current study is interesting with novel and important findings. The authors used a sound methodological approach with adequate results and discussion. Here are some comments for improving the manuscript:

In the abstract:

-In materials and methods section – please mention the number of family members subjected to Sanger sequencing.

-In results section - please replace “Western blotting was showed that….” with “Western blotting has showed that……”

-In conclusion section – please replace “This study show….” with “This study shows……”

In methods:

-P.12 Line 128: The line “It will start in July 2022 and end in December 2023” needs modification and to be in the past tense.

-Please separate the exome sequencing section from the sanger sequencing one.

-Please mention the number of family members subjected to Sanger sequencing and if there any other investigations have been done to them or not.

6. PLOS authors have the option to publish the peer review history of their article (what does this mean?). If published, this will include your full peer review and any attached files.

Reviewer #1: No

Reviewer #2: **Yes: **Hoda Y. Abdallah

---

## [Author Response · Author response to Decision Letter 0]

24 Jul 2024

Dear editors,

Thank you for your letter and for the reviewers’ comments concerning our manuscript entitled “Analysis of low-density lipoprotein receptor gene mutations in a family with familial hypercholesterolemia”(Submission ID number PONE-D-24-12858). Those comments are all valuable and very helpful for revising and improving our manuscript, as well as the important guiding significance to our researches. We have studied comments carefully and have made correction which we hope meet with approval. Revised portion are marked in red in the manuscript. The main corrections in the manuscript and the responds to the reviewer’s comments are listed below this letter.

We would like to express our great appreciation to you and reviewers for comments on our manuscript. If you have any questions, please do not hesitate to contact me.

With best wishes,

Yours sincerely, 

Fan Lin

Jie-Wei Luo

Hong-Li

2024-7-10

Journal Requirements:

Re: Thank you very much for your comments, we have already revised the manuscript and file name according to PLOS ONE's style requirements.

2. Thank you for submitting the above manuscript to PLOS ONE. During our internal evaluation of the manuscript, we found significant text overlap between your submission and previous work in the [introduction, conclusion, etc.].

We would like to make you aware that copying extracts from previous publications, especially outside the methods section, word-for-word is unacceptable. In addition, the reproduction of text from published reports has implications for the copyright that may apply to the publications. Please revise the manuscript to rephrase the duplicated text, cite your sources, and provide details as to how the current manuscript advances on previous work. Please note that further consideration is dependent on the submission of a manuscript that addresses these concerns about the overlap in text with published work. We will carefully review your manuscript upon resubmission and further consideration of the manuscript is dependent on the text overlap being addressed in full. Please ensure that your revision is thorough as failure to address the concerns to our satisfaction may result in your submission not being considered further.

Re: Thank you very much for your comments, we have completely revised the introduction, discussion and conclusion sections in the manuscript.

Re: Sorry, we have corrected the grant numbers and ensured that all information now accurately reflects the funding sources for our study. 

Re: Thank you for your guidance regarding the submission requirements for blot and gel results. We have reviewed PLOS ONE's updated policies on reporting requirements for blots and gels and have prepared our manuscript accordingly.

Re: Thank you for your comments, we have deleted ethics statement in other section besides the Methods.

6. Please include complete captions for your Supporting Information files at the end of your manuscript, and update any in-text citations to match accordingly. Please see our Supporting Information guidelines for more information: http://journals.plos.org/plosone/s/supporting-information.

Re: Thank you for your comments, since we have revised the manuscript, there is currently no supporting information in the manuscript.

Review Comments to the Author

Reviewer 1

The authors reported variants in LDLR gene in a Chinese family with FH. In silico analysis and functional assays were perform. Some major issues need to be addressed.

Major comments:

1.Row 27 (and in whole body of manuscript): LDLR - gene (genes) name should be writen in italics.

Re: Thank you for your comments, we have already revised it in whole body of manuscript.

2.Row 32: Rather genotype-phenotype associations“ than „genetic-phenotypic associations“.

Re: Thank you for your comments, we have already revised it.

3.Rows 49-51: Conclusions“: Presented information are well known for many years and cannot be presented as conclusions of this study. Please, completely revise the text.

Re: Thank you for your comments, we have already revised the conclusions section completely.

4.Row 69: Rather “caused by pathogenic variations” than “caused by variations”.

Re: Thank you for your comments, we have already revised it.

5.Row 77: Rather “common population” than “average people”.

Re: Thank you for your comments, we have already revised it. 

6.Row 86: The author is writing about the apolipoprotein APOB, not the gene, so italics cannot be used.

Re: Thank you for your comments, we have already revised it.

7.Row 93: 1951? Joseph Leonard Goldstein (* 18. April 1940) and Michael Stuart Brown (* 13. April 1941), so at the age of 11 and 10? – reference [17] is from 1974.

Re: Very sorry, the correct statement is 1974, rather than 1951. We apologize for our mistake and have already revised it.

8.Row 176: Use italics for LDLR.

Re: Thank you for your comments, we have already revised it.

9.Row 176: Which “Target genes…”.

Re: Thank you for your comments, the target gene refers to the LDLR gene carrying the c.1027G>A mutation and c.1879G>A mutation.

10.Row 218: …of PROS1? Why PROS1? This is the reason: PMID: 34496879? In my opinion, you overwrote (copy/pasted) your older text.

Re: Very sorry, we accidentally pasted our older text (PMID: 34496879). We apologize for our carelessness and we have already revised it.

11.Row 264: “Simon criteria” are “Simon Broome criteria”, “Dutch Network Diagnostic Criteria” are “Dutch Lipid Clinic Network criteria”.

Re: Thank you for your comments, we have already revised it.

12.Row 301: “According to the… (ACMG)” – there are VCEP criteria for LDLR published in 2022 (PMID: 34906454).

Re: Thank you for your comments, we have already revised it in the manuscript according to the VCEP criteria for LDLR published in 2022.

13. Row 337: Missing space between “Thr” and “mutant”.

Re: Thank you for your comments, we have already revised it.

14.Rows 384 – 566: “Discussion”. This is NOT discussion! Presented information are rather encyclopedic information about LDLR, LDLR protein, LDLR cycle and pathogenic variants in LDLR than discussed results of this study. Why did not you discuss results of assays for Gly343Ser published here PMID: 25647241 or here PMID: 37397863 for Ala627Thr? Please, COMPLETELY revise the text.

Re: Thank you for your comments, we have revised the discussion section completely.

15.Row 520: p.Arg53fs? Why Arg53fs? This is the reason: PMID: 34786791? In my opinion, you overwrote (copy/pasted) your older text.

Re: Very sorry, we accidentally pasted our older text (PMID: 34786791). We apologize for our carelessness and we have already revised it.

16.Row 529: We → we

Re: Thank you for your comments, we have already revised it.

17.Row 548: Severity → severity

Re: Thank you for your comments, we have already revised it.

18.Row 714: Check the reference [39].

Re: Thank you for your comments, due to a complete revision of the discussion, the content citing reference [39] has been deleted.

19.Row 822: Table 1, Head of second column: ago → Age

Re: Thank you for your comments, we have already revised it.

20.Comment: It is necessary to mention that the authors are minimally oriented in the field of LDLR or FH. Errors on line 93 or 264 are clear evidence of this. It is necessary to completely revise the body of the manuscript. Please clearly explain how the errors on Rows 218 and 520 occurred.

Re: Very sorry, we deeply apologize for the oversight of pasting content from our previous publications during the manuscript preparation process. We have made the necessary corrections and have thoroughly revised the introduction, discussion and conclusion sections of the paper.

Reviewer #2: 

The current study is interesting with novel and important findings. The authors used a sound methodological approach with adequate results and discussion. Here are some comments for improving the manuscript:

In the abstract:

-In materials and methods section – please mention the number of family members subjected to Sanger sequencing.

Re: Thank you for your comments, we have mentioned the number of family members subjected to Sanger sequencing in materials and methods section.

-In results section - please replace “Western blotting was showed that….” with “Western blotting has showed that……”

Re: Thank you for your comments, we have already revised it.

-In conclusion section – please replace “This study show….” with “This study shows……”

Re: Thank you for your comments, we have thoroughly revised the conclusion section of the manuscript.

In methods:

-P.12 Line 128: The line “It will start in July 2022 and end in December 2023” needs modification and to be in the past tense.

Re: Thank you for your comments, but the line actually was not mentioned in the manuscript. 

-Please separate the exome sequencing section from the sanger sequencing one.

Re: Thank you for your comments, we have divided the exome sequencing section and the Sanger sequencing section into two parts.

-Please mention the number of family members subjected to Sanger sequencing and if there any other investigations have been done to them or not.

Re: Thank you for your comments, fourteen family members subjected to Sanger sequencing have been mentioned in methods. And all of them performed investigations including routine physical examination, blood biochemistry, electrocardiogram, carotid ultrasound, renal artery ultrasound, echocardiography, and coronary CTA, which has been mentioned in the Research subject section.

---

## [Decision Letter · Decision Letter 1]

6 Aug 2024

PONE-D-24-12858R1Analysis of low-density lipoprotein receptor gene mutations in a family with familial hypercholesterolemiaPLOS ONE

Dear Dr. Luo,

Thank you for submitting your manuscript to PLOS ONE. After careful consideration, we feel that it has merit but does not fully meet PLOS ONE’s publication criteria as it currently stands. Therefore, we invite you to submit a revised version of the manuscript that addresses the points raised during the review process.

We look forward to receiving your revised manuscript.

Kind regards,

Nejat Mahdieh

Academic Editor

PLOS ONE

Journal Requirements:

Reviewers' comments:

Reviewer's Responses to Questions

**Comments to the Author**

1. If the authors have adequately addressed your comments raised in a previous round of review and you feel that this manuscript is now acceptable for publication, you may indicate that here to bypass the “Comments to the Author” section, enter your conflict of interest statement in the “Confidential to Editor” section, and submit your "Accept" recommendation.

Reviewer #1: (No Response)

Reviewer #2: All comments have been addressed

2. Is the manuscript technically sound, and do the data support the conclusions?

Reviewer #1: Yes

Reviewer #2: Yes

3. Has the statistical analysis been performed appropriately and rigorously? 

Reviewer #1: Yes

Reviewer #2: Yes

4. Have the authors made all data underlying the findings in their manuscript fully available?

Reviewer #1: Yes

Reviewer #2: Yes

5. Is the manuscript presented in an intelligible fashion and written in standard English?

Reviewer #1: No

Reviewer #2: Yes

6. Review Comments to the Author

Reviewer #1: The authors reported variants in LDLR gene in a Chinese family with FH. In silico analysis and functional assays were perform. Some minor issues need to be addressed.

Row 104: Who is the proband? Male, 53-year-old Chinese Han? Or Fig 1a, proband = II2, female!?

Row 179 petri dish → Petri dish.

Rows 312 - 313 …have both been → …have been both.

Row 315: …is classified under the PS3 criteria… I do not understand, what author want to say. G1027A is classified as Pathogenic (modified) for Familial Hypercholesterolemia by applying evidence codes (PM2, PP3, PS3, PP4, PS4, PP1_Strong, PM5, PM3, BS4) according to VCEP (https://erepo.clinicalgenome.org/evrepo/ui/interpretation/2fc03ee7-785f-44c9-bac0-10c93ad52227).

Row 316: …is classified as PS4 – the same as Row 315. I do not understand, what author want to say. PS4 (The prevalence of the variant in affected individuals is significantly increased compared to the prevalence in controls) is only one of VCEP criteria met.

Row 337: …Pur, … → …Puro,…

Row 354: Missing space between “Thr” and “mutant”.

Row 369: …of the LDLR. Maybe for clarify … of the mutant (or mutated) LDLR.

Row 382: Rredictprotein → Predictprotein

Row 389: LDLR-deficient disease? Please rephrase this part.

Row 553: In my opinion, Gly + Ser + Ala +Thr is not necessary to clarify.

Row 615: Please check reference 11 and 16 = duplicity.

Row 620: Please check reference 13 and 17 = duplicity.

Row 669: Strom → Strøm.

Reviewer #2: The authors have responded adequately for all the requested comments and the manuscript is suitable for publishing in its current form.

7. PLOS authors have the option to publish the peer review history of their article (what does this mean?). If published, this will include your full peer review and any attached files.

Reviewer #1: No

Reviewer #2: **Yes: **Hoda Y. Abdallah

---

## [Author Response · Author response to Decision Letter 1]

26 Aug 2024

Dear editors,

Thank you for your letter and for the reviewers’ comments concerning our manuscript entitled “Analysis of low-density lipoprotein receptor gene mutations in a family with familial hypercholesterolemia”(Submission ID number PONE-D-24-12858). Those comments are all valuable and very helpful for revising and improving our manuscript, as well as the important guiding significance to our researches. We have studied comments carefully and have made correction which we hope meet with approval. Revised portion are marked in red in the manuscript. The main corrections in the manuscript and the responds to the reviewer’s comments are listed below this letter.

We would like to express our great appreciation to you and reviewers for comments on our manuscript. If you have any questions, please do not hesitate to contact me.

With best wishes,

Yours sincerely, 

Fan Lin

Jie-Wei Luo

Hong-Li

2024-8-17

Journal Requirements:

Re: Thank you very much for your comments, we have already revised the reference list and we have not found any retracted articles among our references. If you identify any, please inform us promptly.

Review Comments to the Author

Reviewer #1

The authors reported variants in LDLR gene in a Chinese family with FH. In silico analysis and functional assays were perform. Some minor issues need to be addressed.

1.Row 104: Who is the proband? Male, 53-year-old Chinese Han? Or Fig 1a, proband = II2, female!?

Re: Very sorry, the proband is indeed II-2 in Fig 1a, a female. We apologize for our mistake and have already revised it.

2.Row 179 petri dish → Petri dish.

Re: Thank you for your comments, we have already revised it.

3.Rows 312 - 313 …have both been → …have been both.

Re: Thank you for your comments, we have already revised it.

4.Row 315: …is classified under the PS3 criteria… I do not understand, what author want to say. G1027A is classified as Pathogenic (modified) for Familial Hypercholesterolemia by applying evidence codes (PM2, PP3, PS3, PP4, PS4, PP1_Strong, PM5, PM3, BS4) according to VCEP (https://erepo.clinicalgenome.org/evrepo/ui/interpretation/2fc03ee7-785f-44c9-bac0-10c93ad52227).

Re: Thank you for your comments, we have already revised it.

5.Row 316: …is classified as PS4 – the same as Row 315. I do not understand, what author want to say. PS4 (The prevalence of the variant in affected individuals is significantly increased compared to the prevalence in controls) is only one of VCEP criteria met.

Re: Thank you for your comments, we have already revised it.

6.Row 337: …Pur, … → …Puro,…

Re: Thank you for your comments, we have already revised it.

7.Row 354: Missing space between “Thr” and “mutant”.

Re: Thank you for your comments, we have already revised it.

8.Row 369: …of the LDLR. Maybe for clarify … of the mutant (or mutated) LDLR.

Re: Thank you for your comments, we have already revised it.

9.Row 382: Rredictprotein → Predictprotein

Re: Thank you for your comments, we have already revised it.

10.Row 389: LDLR-deficient disease? Please rephrase this part.

Re: Thank you for your comments, we have already revised it.

11.Row 553: In my opinion, Gly + Ser + Ala +Thr is not necessary to clarify.

Re: Thank you for your comments, we have already deleted it.

12.Row 615: Please check reference 11 and 16 = duplicity.

Re: Thank you for your comments, we have already revised the reference list.

Row 620: Please check reference 13 and 17 = duplicity.

Re: Thank you for your comments, we have already revised the reference list.

Row 669: Strom → Strøm.

Re: Thank you for your comments, we have already deleted it.

Reviewer #2: The authors have responded adequately for all the requested comments and the manuscript is suitable for publishing in its current form.

Re: Very thank you for your comments.

---

## [Editor Report · Decision Letter 2]

3 Sep 2024

Analysis of low-density lipoprotein receptor gene mutations in a family with familial hypercholesterolemia

PONE-D-24-12858R2

Dear Dr. Luo,

We’re pleased to inform you that your manuscript has been judged scientifically suitable for publication and will be formally accepted for publication once it meets all outstanding technical requirements.

Kind regards,

Nejat Mahdieh

Academic Editor

PLOS ONE
---

## [Editor Report · Acceptance letter]

3 Oct 2024

PONE-D-24-12858R2 

PLOS ONE

Dear Dr. Luo, 

I'm pleased to inform you that your manuscript has been deemed suitable for publication in PLOS ONE. Congratulations! Your manuscript is now being handed over to our production team.

Kind regards, 

on behalf of

Dr. Nejat Mahdieh 

Academic Editor

PLOS ONE